# A focused multi-state model to estimate the pediatric and adolescent HIV epidemic in Thailand, 2005–2025

**Sophie Desmonde**[1,2]*, **Rangsima Lolekha**[3], **Sydney Costantini**[2], **Taweesap Siraprapasiri**[4], **Simone Frank**[2], **Taoufik Bakkali**[5], **Patchara Benjarattanaporn**[6], **Taige Hou**[2], **Supiya Jantaramanee**[7], **Beena Kuttiparambil**[8], **Chuenkamol Sethaputra**[3], **Jeremy Ross**[9], **Andrea Ciaranello**[2,10]

**1** Inserm U1027, Toulouse III University, Toulouse, France, **2** Medical Practice Evaluation Center, Mongan Institute, Massachusetts General Hospital, Boston, Massachusetts, United States of America, **3** Division of Global HIV and TB, U.S. Centers for Disease Control and Prevention, Nonthaburi, Thailand, **4** Department of the Disease Control, Ministry of Public Health, Nonthaburi, Thailand, **5** Joint United Nations Programme on HIV/AIDS (UNAIDS), Regional Support Team, Asia and the Pacific, Bangkok, Thailand, **6** Joint United Nations Programme on HIV/AIDS (UNAIDS), Country Office for Thailand, Bangkok, Thailand, **7** Bureau of Epidemiology, Ministry of Public Health, Nonthaburi, Thailand, **8** UNICEF Thailand Country Office, Bangkok, Thailand, **9** TREAT Asia/amfAR—The Foundation for AIDS Research, Bangkok, Thailand, **10** Division of Infectious Disease, Massachusetts General Hospital, Boston, MA, United States of America

* desmonde.s@chu-toulouse.fr

**Data Availability Statement:** All relevant data are within the manuscript and its Supporting information files.

## Abstract

### Background

We estimated the magnitude of the HIV epidemic among children and youth living with HIV (CYHIV) aged 0–25 years in Thailand, projecting forward from 2005 to 2025, and identified underreported input parameters that influence epidemic projections, in order to inform future public health and research priorities.

### Methods

We developed a focused multi-state transition model incorporating perinatally-acquired HIV and non-perinatally-acquired HIV, stratified by population, including men who have sex with men (MSM), female sex workers (FSW), people who inject drugs (PWID), and the remainder of the population ("other"). We populated the model with published and programmatic data from the Thai national AIDS program when available. We projected the period from 2005–2025 and compared model results to programmatic data and projections from other models. In a scenario analysis, we projected the potential impact of pre-exposure prophylaxis (PrEP) for MSM from 2018–2025.

### Results

The initial 2005 cohort was comprised of 66,900 CYHIV; 8% CYHIV were <5 years, 21% were 5–14 years, and 71% were 15–25 years of age. By 2020, 94% were projected to be >15 years and infections among MSM constituted 83% of all new HIV infections. The numbers of CYHIV decreased over time, projected to reach 30,760 by 2020 (-54%) and 22,640

**Funding:** This work was supported by LIFE+, Vienna, Austria and TREAT Asia-amfAR, the Foundation of Research. AC is funded by the Eunice Kennedy Shriver National Institute of Child Health & Human Development (NICHD) under Award Number R01HD079214. The funders had no role in study design, data collection and analysis, decision to publish, or preparation of the manuscript. The content is solely the responsibility of the authors and does not necessarily represent the official views of the funding entities.

**Competing interests:** The authors have declared that no competing interests exist.

by 2025 (-66%). The proportion of all CYHIV aged 0–25 who were diagnosed and on ART increased from 37 to 60% over the 2005–2025 period. Projections were sensitive to variations in assumptions about initial HIV prevalence and incidence among MSM, PWID, and "other" youth.

## Conclusions

More data on incidence rates among sexual and gender minority youth and PWID are needed to characterize the role of specific exposures and key populations in the adolescent HIV epidemic. More accurate estimates will project shifts in population and inform more targeted interventions to prevent and care for Thai CYHIV.

## Introduction

The Asia-Pacific is the region has the second largest number of people living with HIV (PLHIV) after sub-Saharan Africa, with an estimated 4.7 million people in 2020 [1]. In this year, there were roughly 310,000 children under the age of 14 years and 620,000 youth aged 15–24 years, with young males (53%) slightly outnumbering young females (47%) [2]. The region accounts for almost one in four new infections globally among adolescents aged 10–19 years [3]. In Thailand, there were 470,000 PLHIV by the end of 2019, 3,400 (0.7%) of whom were children aged less than 14 years [4]. Thailand has made substantial progress in preventing HIV and improving care for PLHIV: from 2014 to 2019, the proportion of PLHIV who knew their status increased from 80% to 99.8%, the proportion initiating antiretroviral therapy (ART) and retained in care increased from 51% to 80%, and 97% had virological suppression (<1,000 copies/mL) [4, 5]. Thailand also was the first Asia-Pacific country to reach the milestone of WHO-validated elimination of maternal to child transmission of HIV (MTCT) (<2%) in 2016 [6].

Although Thailand has made progress towards halting the spread of HIV, the total number of HIV infections is rising among adolescents and young adults [3]. Of new infections in children and youth aged 10–24, a substantial proportion occur among men who have sex with men (MSM), young people selling sex, young people who inject drugs, and young transgender people, suggesting that many of these practices conferring risk for HIV begin in adolescence [3, 7]. Furthermore, children who acquired HIV at birth or perinatally in previous decades are now aging into young adulthood, contributing to the population of adolescents living with HIV in Thailand [8]. Young people living with HIV face many challenges, including stigma; lack of effective and tailored support and counseling for HIV testing, treatment, and follow-up; and difficulty with disclosure of their diagnoses to friends and sexual partners [9–12]. Those living with perinatally-acquired HIV face additional challenges associated with family illness and the need to transition from pediatric to adult models of HIV care [13].

Thailand is close to reaching the UNAIDS 95-95-95 targets (95% of PLHIV are aware of their diagnosis, 95% of diagnosed PLHIV are receiving ART, and 95% of PLHIV on ART are virally suppressed) [14]. But treatment coverage and prevention among children and youth living with HIV lags behind adult coverage. To fully reach these goals, there is a need to deliver specific care to children and youth; this requires information about the size and composition of these populations in order to design and implement effective interventions [15–17]. Currently, Thailand's HIV/AIDS estimates for children and youth living with HIV (CYHIV) rely mainly on two published models: the UNAIDS Spectrum software package and the Asia

Epidemic Model (AEM) [18, 19]. Spectrum estimates MTCT rates and HIV prevalence among children aged 0–14. While it also includes youth aged 15–25, it did not until recently distinguish between perinatal and non-perinatal acquisition for people older than 15 [19–21]. AEM is a full-process dynamic transmission model that simulates people older than 15 years [18]. Both models require a complex set of inputs to estimate the numbers of PLHIV and of new infections.

We developed a focused, multi-state model that could be used easily by program planners, with three objectives: 1) to estimate the magnitude of the pediatric and adolescent HIV epidemic among 0-25-year-olds in Thailand between 2005 and 2020, 2) to project the likely trends in CYHIV through 2025, and 3) to inform research priorities by identifying key gaps in available data and models that markedly impact estimates of the pediatric and adolescent HIV epidemic in Thailand.

## Methods

We developed a time-dependent cohort state transition model that models CYHIV between 2005 and 2025 [22]. The model is coded in Python 3.7, implementing health states and transition probabilities shown in Fig 1, with a user interface in Microsoft Excel. Children and youth living with HIV enter the model at age 0 (for infants born to women with HIV) or at ages 13–25 (for all others) and are tracked from time of HIV acquisition through age 25 years plus 364 days or death, whichever occurs first. There are three primary health states: CYHIV who have never been on ART, CYHIV currently on ART, and CYHIV who are lost-to-follow-up (no longer on ART); there are also two absorbing states: death and age >25 years. At the beginning of each yearly cycle, children and youth without HIV face a risk of incident HIV, based on risk group (outlined below). After HIV acquisition, CYHIV transition to the "CYHIV never been on ART" health state. From there, patients face a probability of diagnosis and linkage to care, enabling them to enter the "CYHIV on ART" state. CYHIV on ART in the preceding year also face a probability of becoming lost-to-follow-up (LTFU). Those LTFU in the preceding year have a probability of returning to care. Patients in all health states face state- and age-specific probabilities of mortality. At the end of each cycle, each sub-cohort of children ages up into the next age and year strata (Fig 1).

All probabilities of incident infection and transitions between health states are stratified by calendar year (2005–2025), age (0–25 years), and key population. We modeled six specific and mutually exclusive populations: children and youth with perinatally-acquired HIV (CYPHIV), high-risk MSM, defined as the median proportion of males reporting sex with a male in the last 12 months between 2002–2016, having visited hotspots (Fig 1), low-risk MSM defined as the median proportion of males reporting sex with a male in the last 12 months between 2002–2016 who did not visit hotspots, female sex workers (FSW), people who inject drugs (PWID), and "other." While the first 5 groups are based on parent- or self-reported identifications in the data sources described below the "other" group reflects those who do not self-identify as belonging to any of the populations above and primarily acquired HIV following heterosexual transmission. Because transgender women (TGW) are not consistently defined and reported and are often included within a group reported as MSM or FSW, we chose not to model this group separately [23]. The high-risk to low-risk ratio among MSM was estimated for different provinces, based on a mapping study conducted by the Bangkok Metropolitan Administration in HIV "hotspots" of 8 districts in the capital city of Bangkok. Provinces were assigned high-risk:low-risk ratios based on historical HIV prevalence and incidence data: Bangkok (50:50); Chiang Mai, Phuket, and Chonburi (40:60); and other provinces (30:70). Using the population

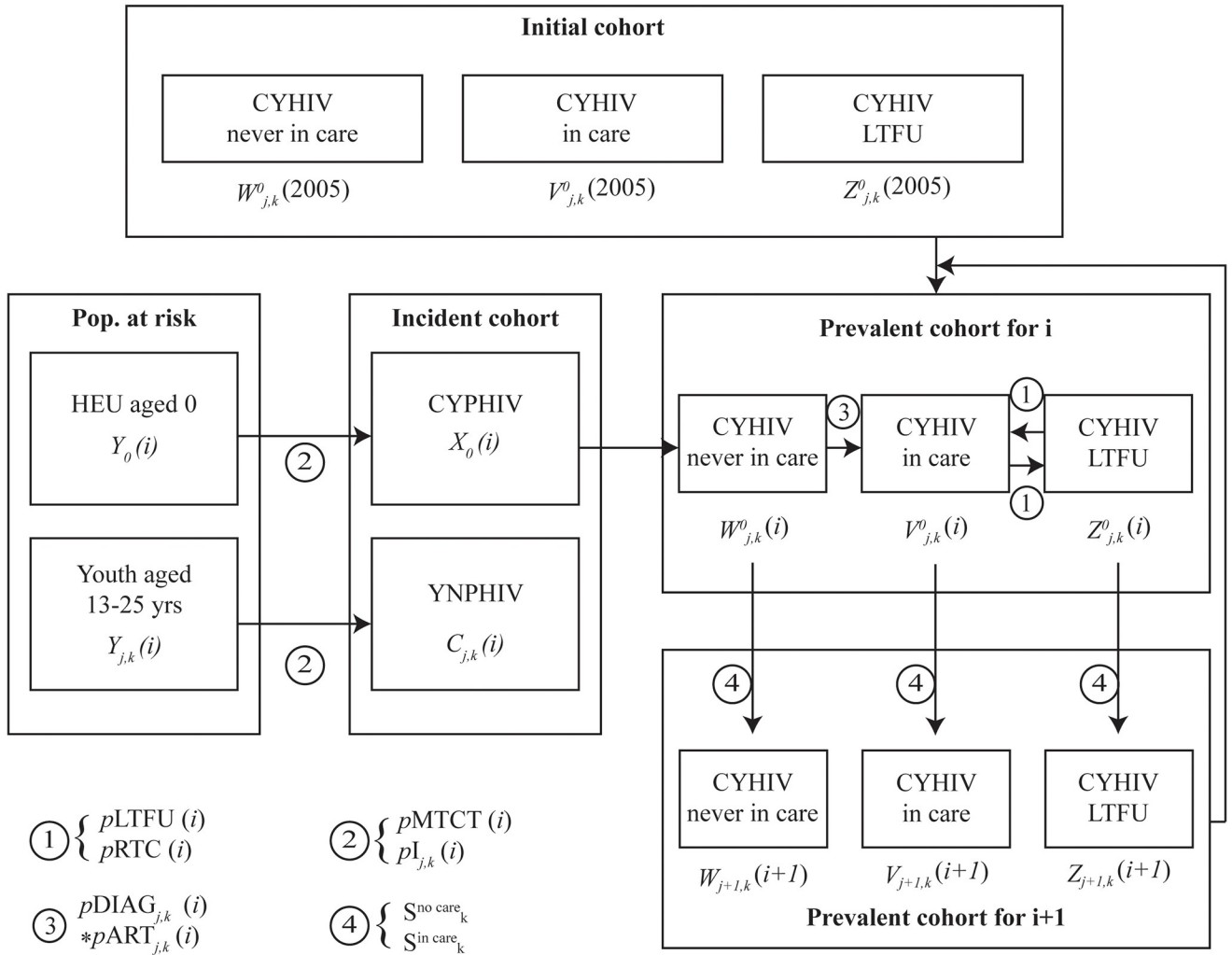

**Fig 1. Model diagram.** (1) At the beginning of each year $i$, those already in care the previous year $V_{j,k}^0$ have a probability of becoming lost-to-follow-up ($p$LTFU($i$)) for each age $j$ and population $k$. Those who were LTFU the previous year $Z_{j,k}$, can return to care ($p$RTC($i$)). (2) Among the population at risk $Y_{j,k}(i)$, newborns have a probability $p$MTCT($i$) to acquire HIV perinatally ($X_0$), and children and youth have a probability $pI_{j,k}(i)$ of acquiring non perinatal HIV ($C_{j,k}$). (3) Those who have never accessed care (new infections and those already known as infected but not in care), $Z_{j,k}$, have a probability of being diagnosed pDIAG$_{j,k}$($i$) and subsequently accessing ART $p$ART$_{j,k}(i)$. (4) Survival rates ($S_k^{care}$ and $S_k^{no\ care}$) are applied to each cohort. Following these steps, children and youth age up into the next year and a new cycle begins. Those who age >25 years exit the model. **CYHIV**: children and youth living with HIV, **HEU**: HIV-exposed, uninfected infants, **CYPHIV**: children and youth living with perinatally-acquired HIV, **YNPHIV**: youth living with non-perinatally-acquired HIV, **MTCT**: mother-to-child transmission, **ART**: anti-retroviral therapy.

size in each of these provinces, we derived a weighted high-risk:low-risk ratio among MSM in Thailand. The uncertainty around this ratio was addressed in sensitivity analyses.

The Thai Pediatric and Adolescent HIV Modeling Working Group was established to review available data, provide recommendations for key input data for this model, and support development of local capacity on the use of the model. The Working Group is comprised of national AIDS program planners, U.S. CDC Thailand office members, clinicians, epidemiologists, and modelers. Model inputs, including all transition probabilities from one state to another were derived for each calendar year, age, sex, and population. Data were derived from national surveys and national projections available at https://hivhub.ddc.moph.go.th/epidemic.php; data prior to 2010 are available upon request. Where national data were not

Table 1. Selected model input parameters.

| Parameter | Value | | | | Source |
|---|---|---|---|---|---|
| **I—Demographic distribution of sub-population data (remainder of 100% are not in any category and not sexually active)** | | | | | |
| | *13-14y* | *15-17y* | *18-19y* | *≥20y* | |
| High-risk MSM* | 0.1% | 0.3% | 0.3% | 0.7% | Derived from BoE. HIV Surveillance System. 2010–2014 [24] |
| Low-risk MSM* | 0.1% | 0.7% | 0.7% | 1.4% | |
| FSW | 0.1% | 0.1% | 0.4% | 0.4% | |
| PWID | 0.1% | 0.1% | 0.1% | 0.1% | Choopanya K, 2013 [25] |
| Other youth | 3.1% | 30.6% | 30.6% | 84.2% | Remainder, weighted by sexual activity, BoE, IBBS 2016 [26] |
| **II—HIV prevalence among those with non-perinatally-acquired HIV in 2005** | | | | | |
| CYPHIV | See S2 Table | | | | |
| High-risk MSM* | 4.8–8.0 | | | | Derived from van Griensven F, 2010 [27] |
| Low-risk MSM* | 3.2–19.2 | | | | Derived from van Griensven F, 2010 [27] |
| FSW | 2.5 | | | | BoE. HIV Surveillance System. 2010–2014, IBBS 2013 [24, 28] |
| PWID | 9.2 | | | | BoE, IBBS 2016 [26] |
| Other youth | 0.1–0.5 | | | | Derived from UNAIDS Data 2017 [29] |
| **III—Incidence rates among those with non-perinatally-acquired HIV (%, yearly)** | | | | | |
| High-risk MSM* | 1.8–4.0 | | | | Derived from van Griensven F, 2015 [30] and van Griensven F, 2018 [31] |
| Low-risk MSM* | 0.7–1.3 | | | | Derived from van Griensven F, 2015 [30] and van Griensven F, 2018 [31] |
| FSW | 0.6 | | | | BoE, IBBS 2016 [26] |
| PWID | 0.7 | | | | Choopanya K, 2013 [25] |
| Other youth | 0.08–0.008 | | | | https://aidsinfo.unaids.org |
| **IV—Access to HIV testing (%, yearly)** | | | | | |
| CYPHIV | 54–94 | | | | GARP indicators |
| High-risk MSM* | 13–29 | | | | National AIDS Program 2017 [32, 33] |
| Low-risk MSM* | 13–29 | | | | P. Chaiphosri, 2016 [33] |
| FSW | 40–60 | | | | National AIDS Program 2017 [32] |
| PWID | 25 | | | | BoE, IBBS 2016 [26] |
| Other youth | 28–40 | | | | Musumari PM, 2016 [34] |
| **V—Access to ART among those tested (yearly)** | | | | | |
| CYPHIV | 52–85 | | | | National AIDS Program 2017 [32] |
| High-risk MSM* | 35–66 | | | | National AIDS Program 2017 [32]. |
| Low-risk MSM* | | | | | P. Chaiphosri, 2016 [33] |
| FSW | 50–67 | | | | National AIDS Program 2017 [32] |
| PWID | 44 | | | | National AIDS Program 2017 [32] |
| Other youth | 18–84 | | | | Derived from UNAIDS-Spectrum estimates |

**MSM**: men who have sex with men, **FSW**: female sex workers, **PWID**: people who inject drugs, **CYPHIV**: children and youth with perinatally-acquired HIV.

* We assigned the proportions designated as high- or low- risk MSM, with high-risk defined as having visited hotspots. These proportions were calculated based on the following hypothesis: in Bangkok (8.5% of the male population), the high-risk:low-risk risk ratio was 1:1; in "tourist" provinces (5.5% of the male population), the ratio was 2:3, and in the rest of the country, the ratio was 3:7. We weighted these proportions by the population size to obtain national estimates by age group, available in S1 Table.

available, we derived model inputs from clinical trials and cohort studies in Thailand or other settings (Table 1, S1–S3 Tables, S1 and S2 Figs).

Data were available to inform all model input parameters through calendar year 2018. For years 2019–2025, we assumed that the following parameters would remain equal to their 2018 values: demographic distribution of each key population; new perinatal HIV infections; and, for each key population, incidence rates among those at risk for NPHIV, access to HIV testing,

ART, LTFU, return to care, and survival. The demographic structure of the Thai population aged 0–25 years, available from the World Population Prospects 2017 Revisions, was the only varying input in the modeled 2019–2025 period [35].

Further methodological details are provided in S1 Appendix.

## Comparison with existing models

We compared output from this model to results from the AEM-Spectrum model (https://aidsinfo.unaids.org). Parameters generated by both models, and thus able to be compared, included total number of children, adolescents, and youth aged 0–14 years, 10–19 years, and 15–24 years living with HIV in the 2005–2019 period and number of new infections in each of these age groups in the same period. We were unable to compare directly to AEM projections as AEM results are reported in aggregate for all people aged 15–49, and a breakdown of AEM results for youth aged 15–24 years was not available [18].

## Sensitivity analyses

We evaluated the impact of uncertainty in key model input parameters on our results by conducting extensive one-way and multi-way sensitivity analyses. Key parameters included prevalence of HIV among key populations in 2005, high-risk:low-risk ratio among MSM, and HIV testing coverage among MSM and FSW. For each population, we conducted model runs using a range of values derived from published reports.

## Scenario analyses: Pre-exposure prophylaxis (PrEP) scale-up

In 2018, the Government of Thailand recommended pre-exposure prophylaxis (PrEP) for MSM as an additional HIV prevention program within Thailand's National Guidelines on HIV/AIDS Treatment and Prevention. To inform the potential implementation of PrEP, we projected the number of new infections among MSM aged 13–25 if PrEP were provided starting in 2018 and continued through 2025. We evaluated several different scale-up scenarios based on previously published uptake studies of PrEP in the MSM population in Thailand (assuming 41% uptake) and the US (assuming 75% uptake), as well as an ideal 100% uptake scenario [36, 37]. For MSM receiving PrEP in these scenarios, we used an incidence rate reduction of 49% [25, 38]. We assumed that PrEP uptake and incidence rate reduction was the same for all age groups and among both high- and low-risk MSM, and that uptake of PrEP would remain constant from 2018 onward (S4 Table).

## Dissemination and training

The tool input sheets, output template and Python code are publicly available at https://github.com/MGH-MPEC/Thailand-Model. Furthermore, the Thai Pediatric and Adolescent HIV Modeling Working Group identified model users within their institutions who learned to use the model through a series of ten live training session.

## Ethics statement

All data were aggregated and no individual-level human subjects' data were used: there was no requirement for informed consent. This study was reviewed by the Mass General Brigham IRB and approved as "not human subjects research."

**Table 2. Projected number of children and youth aged 0–25 years living with HIV and new HIV infections in selected years.**

|  | 2005 | 2010 | 2015 | 2020 | 2025 |
|---|---|---|---|---|---|
| **Children and youth living with HIV (total)** [1] | | | | | |
| **CYPHIV** | 19,080 | 17,040 | 15,990 | 12,290 | 6,930 |
| **Overall MSM** | 9,490 | 13,630 | 14,770 | 14,360 | 13,350 |
| *High-risk MSM* | 5,250 | 7,780 | 8,430 | 8,220 | 7,640 |
| *Low-risk MSM* | 4,230 | 5,850 | 6,340 | 6,150 | 5,710 |
| **FSW** | 1,000 | 1,090 | 880 | 820 | 770 |
| **PWID** | 1,560 | 1,100 | 770 | 570 | 540 |
| **Other youth** | 35,820 | 19,220 | 9,580 | 2,710 | 1,050 |
| **Total** | 66,940 | 52,070 | 41,980 | 30,760 | 22,640 |
| **Number of new HIV infections per year** [1] | | | | | |
| **CYPHIV** | 410 | 180 | 70 | 70 | 70 |
| **Overall MSM** | 2,810 | 3,310 | 2,800 | 2,840 | 2,630 |
| *High-risk MSM* | 1,630 | 1,890 | 1,590 | 1,610 | 1,490 |
| *Low-risk MSM* | 1,190 | 1,420 | 1,210 | 1,230 | 1,130 |
| **FSW** | 190 | 180 | 190 | 180 | 170 |
| **PWID** | 100 | 100 | 100 | 100 | 90 |
| **Other youth** | 5,520 | 2,670 | 1,320 | 240 | 220 |
| **Total** | 9,030 | 6,440 | 4,480 | 3,430 | 3,180 |

[1] Values are rounded to the nearest 10. Total values may not add up due to rounding. Model results are based on empiric data through 2020 where available and projected forward assuming minimal change from 2018 data through 2025.

## Results

### Total and new infections over time, 2005–2025

In 2005, we estimated that 66,940 children, adolescents, and youth aged 0–25 years were living with HIV. Of these CYHIV in 2005, 19,080 (30%) acquired HIV perinatally, 9,490 (14%) were high- or low-risk MSM, 1,000 (1%) were FSW, and 1,560 (2%) were PWID (Table 2, S3 Fig). The number of model-estimated CYHIV in Thailand decreased steadily over time, reaching 30,760 by 2020 (-54%). Estimated new HIV infections each year also decreased, from 9,030 in 2005 to 3,430 in 2020. Projecting forward in time, there will be a total of 22,640 CYHIV anticipated in 2025 (-26% since 2020), and 3,180 new infections among 0-25-year-olds in 2025.

### Distribution by age

Youth aged 15–25 represented the largest proportion of all CYHIV in 2005, comprising about 71%. We estimated a steady aging of the CYHIV population over time, with the 15-25-year-old group growing to include 95% of the CYHIV population by 2020 and ≥95% by 2025 (Fig 2). Children <5 years old comprised a small proportion (8%) of the 0-25-year-old population in 2005, decreasing to <2% by 2025.

### Distribution by risk group

In 2005, we estimated that there were 19,080 CYPHIV. Between 2005 and 2020, the number of estimated new perinatal HIV infections decreased by 82%, dropping from 410 to 70 per year. As the number of new perinatal infections decreased over time, we observed a shift in the age of the CYPHIV population: while 28% were aged <5 years in 2005, by 2020, 3% were <5 years, and 88% were projected to be in the 15–25 age bracket (S4 Fig). CYNPHIV were

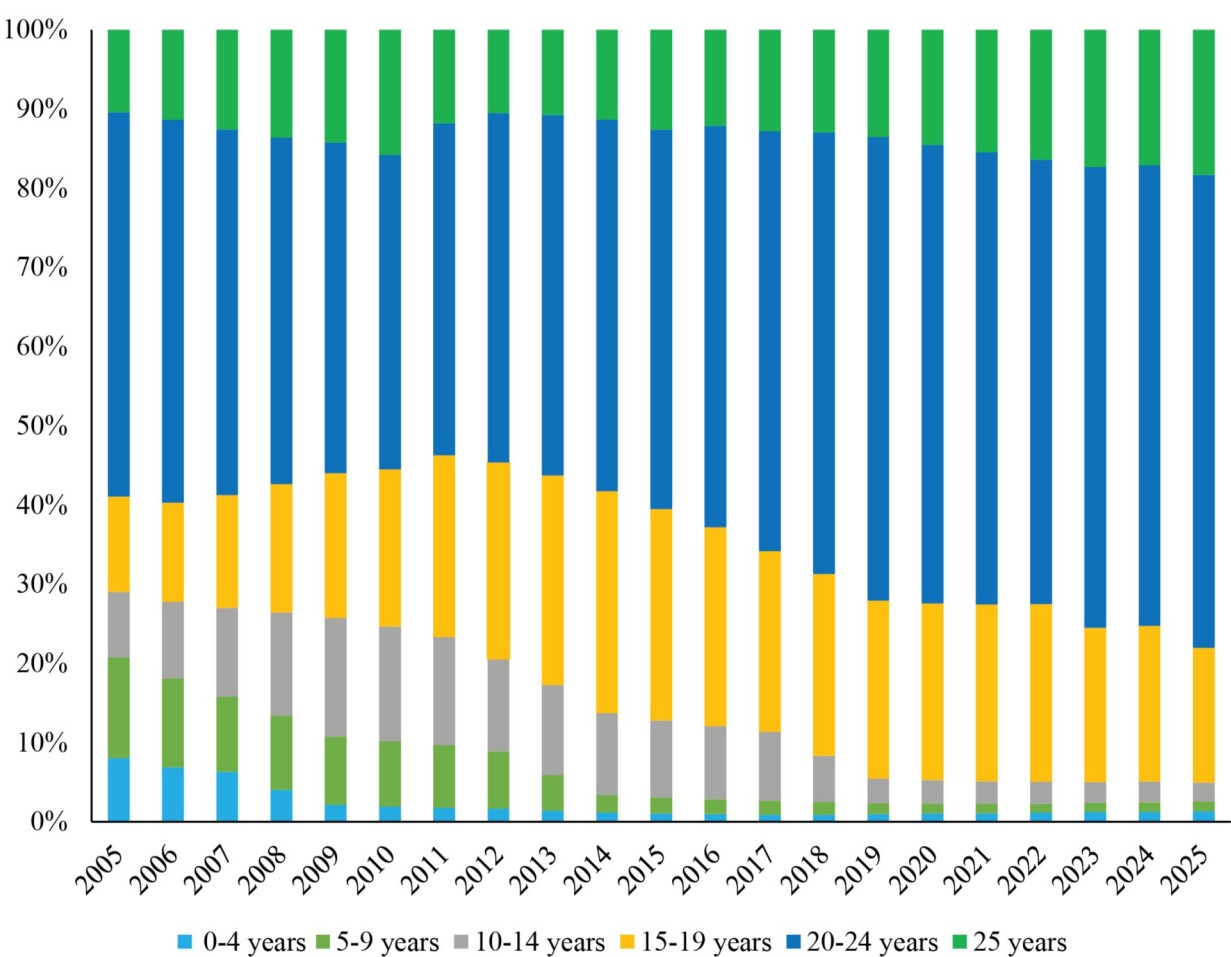

**Fig 2. Model-projected distribution of children and youth living with HIV by age group, 2005–2025.** The vertical axis shows the proportion of all CYHIV in Thailand who fall into each age category in each modeled calendar year. The horizontal axis shows each calendar year from 2005 to 2025. Children and youth aged 0–4, 5–9, 10–14, 15–19, 20–24, and 25 years are shown in different shades of blue and green.

estimated to constitute the vast majority of CYHIV aged 0–25. By 2025, we projected CYNPHIV would account for 15,710 CYHIV (69%); this total included 3,100 new non-perinatal infections acquired in 2025. We projected a large decrease in the number of new infections among "other youth" aged 0–25 over time, from 5,520 in 2005 to 220 in 2025. The numbers of CYHIV in all other populations except MSM also declined over time, although to a lesser degree (Table 2, S5 Fig).

The largest number of both new and total HIV infections among adolescents and youth in Thailand was projected to occur among young MSM, for whom the number of new infections remained steady around ~2,800 between 2005–2025 (Table 2). This included both high-risk and low-risk MSM, as defined in Methods. For the year 2020, we projected that 14,360 young MSM were living with HIV (both high- and low-risk), representing 47% of the overall population of CYHIV. Although the number of young MSM living with HIV was projected to decline over time, reaching 13,350 by 2025, the proportion of MSM among CYHIV was projected to increase to 59% by 2025. By 2025, new infections among MSM (high- and low-risk) were estimated to account for 83% of new infections in this overall age group.

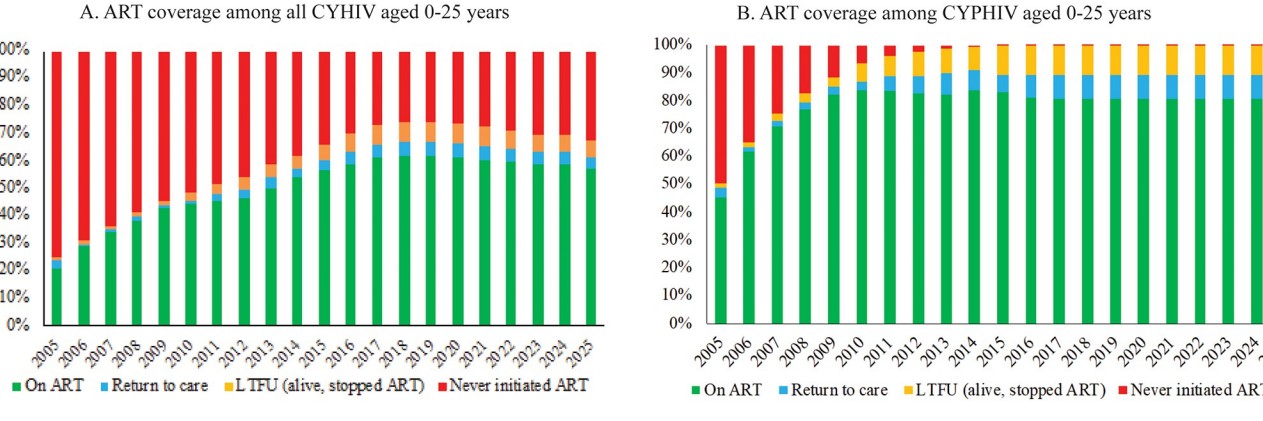

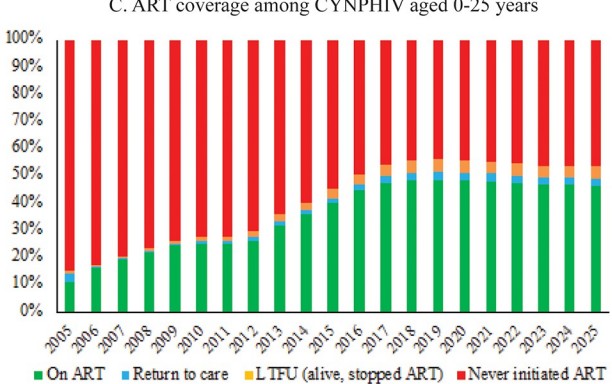

**Fig 3. ART coverage among CYHIV aged 0–25 years in Thailand, 2005–2025.** The vertical axis shows the proportion of all CYHIV aged 0–25 years in Thailand who fall into each category of ART coverage: on ART (green), returned to care (blue), stopped ART (yellow), and never initiated ART (red). The horizontal axis shows each calendar year from 2005 to 2025. Each panel represents either all CYHIV (A), only CYPHIV (B), or only CYNPHIV at the age of 13–25 years (C). **ART**: antiretroviral therapy, **CYHIV**: children and youth living with HIV, **CYPHIV**: children and youth living with perinatally acquired HIV, **CYNPHIV**: children and youth living with non-perinatally acquired HIV.

## ART coverage

The proportion of all CYHIV aged 0–25 who were projected to be diagnosed and on ART ranged from 21% in 2005 to 61% in 2020 (Fig 3 and S5 Table). In forward projections, we projected a slight decrease in ART coverage (to 57%) by 2025, as CYHIV dropped out of care (Fig 3A). Despite this, the projected absolute numbers of CYHIV who never initiated ART or who stopped ART continued to decrease over time between 2005–2025 (Fig 3). We estimated that 45% of CYPHIV ≤ 25 years were on ART in 2005. By 2009, ART coverage met the then-current 90-90-90 UNAIDS target for CYPHIV diagnosed and on ART in 2009 (at 82% of total CYPHIV, exceeding 90%*90% = 81%). ART coverage reached a maximum of 84% in 2014 and then dropped to 81% in 2015, as children and youth dropped out of care (Fig 3B). Of CYHIV living with non perinatally-acquired HIV and aged 13–25 years, ART coverage was 11% in 2005 increased over time, reaching 46% by 2025 (Fig 3C).

## Sensitivity analyses

Sensitivity analyses results are available in S6 Table. We identified two key gaps in data in input data, affecting model estimates. We found results were sensitive to variation in the MSM

high-risk: low-risk ratio and initial prevalence of HIV in 2005 among each population at risk for non-perinatally acquired HIV. As we increased the proportion of high-risk MSM, the number of new MSM infections also increased. When increasing the prevalence of HIV in 2005 by 2-fold in each risk group, we projected higher numbers of CYNPHIV, however, towards the later years, as those living with HIV in 2005 aged out of the model, our projections then matched the base-case. Results were least sensitive to variations in HIV testing and ART coverage among MSM, although increased HIV testing and ART coverage in later years resulted in higher projected numbers of MSM living with HIV, as their survival increased.

### Comparison with AEM-spectrum model projections

Between 2005–2011, we projected higher numbers of CYHIV aged 0–14 years than were estimated by the AEM-Spectrum model for the same period (S6 Fig). This trend then reversed after 2011, as older CYHIV aged out of the model, and our model projected smaller numbers of children aged 0–14 years living with HIV. When comparing number of new infections, we found a lower number of new perinatal infections before 2017 (S7 Fig). Beyond 2017, our model estimates and AEM-Spectrum estimates were similar.

Additionally, we compared estimates to AEM-Spectrum outputs specific to 10-19-year-olds and 15-24-year-olds (S8 and S9 Figs). For adolescents aged 10–19 years, we report a higher number of youths living with HIV compared to AEM-Spectrum in the 2005–2015 period. In later projections, 2015–2019, estimates were of the same order of magnitude in both models. When restricting to CYPHIV (S8 Fig), we found a similar rate of decline in the number of 10-19-year-olds living with HIV as was projected by AEM-Spectrum. However, our estimates of absolute numbers of CYPHIV aged 10–19 (S8 Fig, difference between orange and purple lines) were 2-fold lower than the AEM-Spectrum outputs. Among 15-24-year-olds (S9 Fig), our model also projected substantially lower numbers of CYHIV compared to AEM-Spectrum in early calendar years, becoming similar by 2013. Then, as data became more available, our model projections were similar to those projected by AEM-Spectrum (S10 and S11 Figs).

We found that our model projected comparable estimates to those of AEM-Spectrum in all calendar years only when we assumed larger numbers of CYHIV entering the model in 2005 (2-fold greater HIV prevalence in 2005 among MSM, FSW, PWID, and other youth aged 15–24 years). With this assumption, however, our model still estimated larger numbers of CYHIV aged 10–19 compared to AEM-Spectrum (S12 and S13 Figs).

When increasing access to HIV testing among adolescent MSM and FSW in 2019 to match the UNAIDS indicator for awareness of HIV status in these adult key populations, we found ART coverage among all CYHIV increased from 46% to 59%, but remained below the UNAIDS target of 95% (S14 Fig).

### Scenario analyses: PrEP scale-up

In this example, we set the introduction of PrEP in 2018 and found the number of new infections among young MSM was projected to be markedly lower, decreasing by 17–49% depending on uptake (S15 Fig).

### Discussion

We developed a focused, multi-state transition model to estimate the magnitude of the pediatric and adolescent HIV epidemic among 0-25-year-olds in Thailand between 2005 and 2025 and to identify key gaps in input data to inform research and intervention priorities. Our analysis had several main findings. First, we estimated a total of 30,760 children, adolescents, and youth aged 0–25 years are living with HIV in Thailand in 2020, among whom one-third are

living with perinatally-acquired HIV. Second, in the 2018–2025 period, we estimated high numbers of young MSM living with HIV who comprise a large proportion of the adolescent and youth HIV epidemic in Thailand; this highlights urgent needs to identify young MSM at risk for HIV and to increase their access to HIV testing and treatment. Third, we projected a slight decrease in ART coverage as CYHIV dropped out of care; lack of ART coverage was most pronounced in CYNPHIV, for whom ART coverage remained <50%. Fourth, our estimates were sensitive to variation in key model input parameters, such as prevalence of HIV in 2005 and incidence in later calendar years, underscoring the need for more complete and robust data on HIV in adolescents and youth.

Our results emphasize both the successes and long-term implications of PMTCT programs in Thailand. We estimated a steady decline in perinatal HIV infections in Thailand, reflecting the implementation of new national guidelines for PMTCT of HIV (WHO Option B in 2010 and Option B+ in 2014) as part of a country-wide effort to eliminate pediatric HIV [6, 39, 40]. Furthermore, we projected ART coverage among CYPHIV to remain >90% since 2014, reflecting implementation in 2007 of an active case management network to identify infants with HIV and in 2014 of a program to accelerate ART initiation after diagnosis [6, 41]. Access to care and high survival rates mean that CYPHIV are now estimated to constitute a third of all CYHIV in Thailand. Furthermore, with fewer new infections and a longer survival, we observe an aging CYPHIV population and a reduction of the proportion on ART as the proportion of those dropping out of care increased. As more CYPHIV live longer, it is important to address through interventions and policy changes the new challenges that this aging population may face, such as transition to adult care, adherence to treatment, and retention in care [8, 15].

Our model results also suggest that the adolescent HIV epidemic in Thailand is now driven mainly by adolescents and youth living with non-perinatally-acquired HIV, particularly MSM. Most new HIV infections among children and youth are among MSM aged 13–24, comprising 65% of all new infections by 2025. These results are in line with an observational cohort study reporting steady high HIV incidence among young MSM in Bangkok while the HIV incidence among MSM is declining [42]. However, our findings depend on data about HIV incidence rates among young MSM, and representative data on HIV incidence are limited in Thailand. In a systematic review conducted in 2013 on HIV incidence, authors reported 53 studies in Thailand since 2005, of which only 5 were conducted among MSM [43]. Furthermore, each of these studies took place in Bangkok, which may not be fully representative as MSM communities emerge in other cities [27, 30, 43–45]. Our use of these specific HIV incidence rates may have overestimated the HIV epidemic among MSM aged 13–24 years. Although better data on HIV incidence among young MSM are clearly needed, the large estimated burden of the CYHIV epidemic in Thailand borne by MSM is consistent with previously described findings and suggests the need for more targeted interventions among this group [45]. In scenario analyses, we found that PrEP among MSM is likely to reduce the total number of CYHIV in Thailand (by 17–49% among MSM, and by 12–29% among all CYHIV, depending on PrEP uptake). In these analyses, we used a 49% incidence rate reduction, reported from an adult PWID population in Thailand, and in line with previously reported data from the iPrEX study conducted among an MSM population in the USA [25, 46]. Additionally, recent data has reported high uptake and efficacy of PrEP among MSM in Thailand, reaching a 75% risk reduction, suggesting that the scale-up of PrEP in Thailand among high and low risk MSM maybe have even higher impact on the number of new infections among children and youth overall [38].

In addition to data specific to MSM, more data are also needed on TGW, PWID, and FSW to elucidate the role these risk groups play in the adolescent HIV epidemic. These populations

are not mutually exclusive, and they are not consistently defined in currently available data. Additionally, the burden of drug use is high among young MSM and FSW populations and data available to date do not specifically address potential overlap between groups identifying as PWID, MSM, and FSW [47–49]. Consequently, our model may overestimate the adolescent and youth epidemic when high-risk populations overlap. Furthermore, inputs related to access to care were estimated from the National AIDS Programme (NAP) database. Although the care provider can categorize key population (MSM, PWID, FSW. . .) in the database, this variable is often missing. In our analyses, we made several assumptions based on data available from IBBS to stratify access to care by population. Improving the completeness of population-categorized data entry in NAP would be helpful for monitoring each population's HIV cascade, providing HIV targeted services for specific populations as needed, and improving data report that can be used for modeling and projections in the future.

We projected that both PWID and FSW constitute a small proportion of CYHIV aged 13–24 between 2005 and 2025 (<5%). Data about HIV risk among young FSW since 2005 are scarce, and this population is often excluded from population-based surveys and model projections. However, recent survey data from the Asia-Pacific region has revealed that a high percentage of FSW begin engaging in transactional sex in their early teens [50]. While PWID remain a small proportion of the adolescent HIV epidemic in Thailand, the number of HIV infections remains high within this population. HIV prevention campaigns focusing on PWID need to be adolescent-friendly in order to further contain the HIV epidemic in Thailand.

We projected a large decrease in the number of new infections among "other youth" aged 0–25 over time. This can explain both by an increase in roll-out and uptake of prevention intervention for heterosexual HIV transmission over time with increasing access to sexual reproductive health services, and the ageing out of the model of those acquiring HIV in the earlier years.

Overall, ART coverage was projected to reach 57% by 2025, lagging behind that of and significantly below the 95-95-95 targets and emphasizing the specific needs of this pediatric population. Furthermore, while projected ART coverage among CYPHIV to remain >90%, it was considerably among those living with non-perinatally acquired HIV was significantly lower compared to those living with perinatally-acquired HIV, reaching <50%, significantly below the 95-95-95 targets. In context where the epidemic is driven mainly by key populations, this underlines the needs to population and age specific interventions to increase ART access and improve health outcomes in this population.

We compared our model outputs to the AEM-Spectrum outputs and found comparable estimates for those living with perinatally-acquired HIV aged 0–14 years. However, in older adolescents and youth, our model estimates were substantially different than AEM-Spectrum estimates. Among adolescents aged 10–19 years, while estimates for 2015–2025 were comparable, our model projected significantly larger estimates for the early (2005–2015) period. This pattern was mainly driven by CYPHIV and can be explained by the high testing and ART coverage rates among CYPHIV entering the model in 2005 who survived until 2015 before aging out of the model. There is the need for additional characterization of these surviving CYPHIV >19 years as they enter adulthood. Among 15-24-year-olds, our estimates of total number of CYHIV were also lower than AEM-Spectrum outputs. In sensitivity analyses, when we increased HIV prevalence in 2005 among older youth 2-fold, we found more comparable estimates for this age group. Furthermore, our model estimated a comparable number of new infections in all calendar years to those estimated by AEM-Spectrum model.

There are several limitations to our analysis. First, our model does not track individual patient trajectories over time; however, « **memory** » **of previous health states is not essential to the projections made in this analysis, and the simpler compartmental model structure**

allows for simple and transparent sensitivity analyses which can identify key parameters influencing model outputs. **Second, we do not include estimates of uncertainty in our base-case model results. Instead, we conduct extensive sensitivity analyses to quantify the impact of uncertain in key parameters; we found that our model was mainly sensitive to variations in the parameters defining the MSM population (high:low risk ratio, access to diagnosis and care) and the initial HIV incidence in 2005 in each group**. Third, we assumed exclusive formula feeding in all infants > 1 year born to mothers living with HIV, **because breastfeeding beyond this age is rare among women living with HIV in Thailand. Given low maternal prevalence (<1%), and MTCT rate (<2%), even large relative changes in vertical transmission risk after age 1 would have minimal impact on our projections, and CYPHIV would remain a small proportion of the projected epidemic by 2025**. Fourth, our analysis is limited by gaps in available data for use as input parameters. In particular, by carrying most recent available inputs through to 2025 in our base case analysis, we have not **simulated potentially** improved care and interventions in the child and adolescent population, **including improved data reporting and completeness of data over time; at the same time, we did not** account for the impact of the COVID-19 pandemic on access to care. These conservative inputs however, provide the reader with a projection of the epidemic in a *status quo* scenario. **As post-COVID services resume, and if access to HIV diagnosis and care improves over time, ART coverage would be higher than that estimated and number of new infections lower. Fifth, we deliberately do not model HIV transmission between individuals, due to lack of relevant data on sexual and injection networks in Thai youth; we chose instead to develop a simpler model structure, using readily available data about yearly probabilty of HIV acquisition in each cohort, to identify the most relevant parameters and inform future data collection**. Finally, although our model does not directly simulate the impact of prevention strategies on transmission, incidence rates can be adjusted to account for an intervention by age, year, and risk group, as was demonstrated with PrEP.

Taken together, our findings highlight that better data about HIV prevalence and incidence among each population group, as well as about the proportion of youth who are part of each group, are critical. Lack of consistent reporting, in particular of sexual and gender diversities among adolescents, impairs the ability to quantify and address HIV risk among the global adolescent populations, as the needs of these youth are left unmet. Extensive sensitivity analyses and comparison with other available estimates have allowed us to identify the most important data still needed. Additionally, frequency of risk behaviors in youth, HIV incidence in each risk group, and survival in non-perinatally infected youth are key inputs.

## Conclusion

We estimate that while Thailand has achieved the elimination of vertical transmission, there remains a substantial population of youth >15 years living with perinatally-acquired HIV. In addition, the adolescent epidemic is driven predominantly by high rates of HIV among adolescent and youth who identify as MSM. Our model is the first to specifically focus on and characterize the adolescent population in Thailand. A focused model, in comparison with other available estimates, provides a unique opportunity to identify critical data gaps and inform future research.

## Supporting information

**S1 Appendix.**
(DOCX)

**S1 Table. Model input parameters (including those shown in manuscript Table 1).**
(DOCX)

**S2 Table. Number of children aged 0–15 years living with HIV in 2005 (results from adjacent tool used to derive model inputs).**
(DOCX)

**S3 Table. Derivations of the overall mother-to-child-transmission rates.**
(DOCX)

**S4 Table. Summary of incidence inputs for scenario analyses on PrEP scale-up among MSM.**
(DOCX)

**S5 Table. Model-projected number of children and adolescents living with HIV according to care status.**
(DOCX)

**S6 Table. Summary of sensitivity analyses.**
(DOCX)

**S1 Fig. Model-projected versus PHIMS-reported mother-to-child-transmission rates, 2005–2015.**
(TIF)

**S2 Fig. Model-projected versus reported proportion of CYHIV on ART among those who ever initiated treatment, 2008–2016.**
(TIF)

**S3 Fig. Model-projected number of children and youth living with HIV by risk group, 2005–2025.**
(TIF)

**S4 Fig. Model-projected age distribution among children and youth living with perinatally-acquire HIV, 2005–2025.**
(TIF)

**S5 Fig. Number of new HIV infections among children and youth living with non-perinatally-acquired HIV, 2005–2025.**
(TIF)

**S6 Fig. Number of children aged 0–14 years living with HIV in Thailand, 2005–2019—Comparison with Spectrum output.**
(TIF)

**S7 Fig. Number of new infections among children aged 0–14 years in Thailand 2005–2019—Comparison with Spectrum output.**
(TIF)

**S8 Fig. Number of children aged 10–19 years living with HIV in Thailand 2005–2019—Comparison with Spectrum output.**
(TIF)

**S9 Fig. Number of children aged 15–24 years living with HIV in Thailand 2005–2019—Comparison with Spectrum output.**
(TIF)

**S10 Fig. Number of new infections among adolescents aged 10–19 years in Thailand 2005–2019—Comparison with Spectrum output.**
(TIF)

**S11 Fig. Number of new infections among adolescents aged 15–24 years in Thailand 2005–2019—Comparison with Spectrum output.**
(TIF)

**S12 Fig. Sensitivity analysis: Number of children aged 10–19 years living with HIV in Thailand 2005–2019—Comparison with Spectrum output.** In this sensitivity analysis, the prevalence of HIV in 2005 is doubled compared to the base case.
(TIF)

**S13 Fig. Sensitivity analysis: number of youth aged 15–24 years living with HIV in Thailand 2005–2019—Comparison with Spectrum output.** In this sensitivity analysis, the prevalence of HIV in 2005 is doubled compared to the base case.
(TIF)

**S14 Fig. ART coverage among CYHIV aged 0–25 years in Thailand, when increasing access to HIV testing among MSM and FSW, 2005–2025.** The vertical axis shows the proportion of all CYHIV aged 0–25 years in Thailand who fall into each category of ART coverage (on ART, returned to care, stopped ART, and never initiated ART). The horizontal axis shows each calendar year from 2005 to 2025. **ART**: antiretroviral therapy, **CYHIV**: children and youth living with HIV, **CYPHIV**: children and youth living with perinatally acquired HIV, **CYNPHIV**: children and youth living with non-perinatally acquired HIV.
(TIF)

**S15 Fig. Number of people living with HIV among MSM aged 13–25 years in different PrEP uptake scenarios, 2015–20.**
(TIF)

## Acknowledgments

We thank the CEPAC research team at the Medical Practice Evaluation Center at Massachusetts General Hospital and Dr. Annette Sohn of TREAT Asia for facilitating this analysis. We are particularly grateful to Giulia Park, Christopher Alba, and Alyssa Amick for assistance with manuscript preparation. Finally, we would like to acknowledge with gratitude, valuable discussion and feedback—Dr. Eileen Dunne, Ms. Thananda Naiwatanakul, Dr. Suchunya Aungkulanon, Dr. Sarika Pattanasin, Dr. Marta Ackers, Dr. Michael Martin, Ms. Suvimon Tanpradech of Thailand MOPH-U.S. CDC Collaboration (TUC); Dr. Thanyawee Puthanakit, Dr. Wipaporn Songtaweesin of Chulalongkron University; Dr. Walairat Chaifoo of Bureau of AIDS TB and STIs (BATS); Dr. Wadchara Pumpradit of Bumrungrad International Hospital; Dr. Kulkanya Chokephaibulkit, Dr. Supattra Rungmaitree of Siriraj Hospital, Mahidol university; Dr. Ye Yu SHWE of UNAIDS; Ms. Pornthip Khemngern, Division of AIDS and Sexually Transmitted Infections, Department of Disease Control, Thailand's Ministry of Public Health; Ms. Niramon Punsuwan of Bureau of Epidemiology, Thailand's Ministry of Public Health.

## Author Contributions

**Conceptualization:** Sophie Desmonde, Rangsima Lolekha, Sydney Costantini, Simone Frank, Andrea Ciaranello.

**Data curation:** Rangsima Lolekha, Taweesap Siraprapasiri, Taoufik Bakkali, Patchara Benjarattanaporn, Supiya Jantaramanee, Beena Kuttiparambil, Chuenkamol Sethaputra, Jeremy Ross.

**Formal analysis:** Sophie Desmonde, Sydney Costantini.

**Methodology:** Sophie Desmonde, Simone Frank, Taige Hou.

**Software:** Taige Hou.

**Supervision:** Jeremy Ross, Andrea Ciaranello.

**Validation:** Sophie Desmonde.

**Writing – original draft:** Sophie Desmonde.

**Writing – review & editing:** Rangsima Lolekha, Sydney Costantini, Taweesap Siraprapasiri, Simone Frank, Taoufik Bakkali, Patchara Benjarattanaporn, Taige Hou, Supiya Jantaramanee, Beena Kuttiparambil, Chuenkamol Sethaputra, Jeremy Ross, Andrea Ciaranello.

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
