## [Decision Letter · Decision Letter 0]

8 Jun 2022

PONE-D-21-40026A focused multi-state model to estimate the pediatric and adolescent HIV epidemic in Thailand, 2005 – 2025PLOS ONE

Dear Dr. Desmonde,

Thank you for submitting your manuscript to PLOS ONE. After careful consideration, we feel that it has merit but does not fully meet PLOS ONE’s publication criteria as it currently stands. Therefore, we invite you to submit a revised version of the manuscript that addresses the points raised during the review process.

ACADEMIC EDITOR:

Your paper was of interest to the reviewers. As you can see below, the reviewers identified both strengths and weaknesses in your paper. The reviewers noted important areas of your paper that require careful attention. From my own reading of your paper, I am in agreement with the reviewers that your paper will make an important contribution and that your paper will benefit from a minor revision. Please revise your paper in accordance with the reviews. In addition to the reviewer's comments please address the following in your revised manuscript.

1. Please review figures and tables - Table 2 has text cut off in the second column. The text in the figures are difficult to read. The display quality of the figures are poor - please increase the resolution of the figures.

2. Please include in the limitation the impact on the major assumptions in the model and how that likely affects the results. 

Please be sure to include a cover letter with your revision that addresses all of the reviews and comments as well as my comments, point-by-point. Your revision should also carefully follow the journal style as presented in the Instructions for Authors available at the journal website.  

We look forward to receiving your revised manuscript.

Kind regards,

Brian C. Zanoni, MD

Academic Editor

PLOS ONE

Journal Requirements:

Additional Editor Comments (if provided):

Your paper was of interest to the reviewers. As you can see below, the reviewers identified both strengths and weaknesses in your paper. The reviewers noted important areas of your paper that require careful attention. From my own reading of your paper, I am in agreement with the reviewers that your paper will make an important contribution and that your paper will benefit from a minor revision. Please revise your paper in accordance with the reviews. In addition to the reviewer's comments please address the following in your revised manuscript.

1. Please review figures and tables - Table 2 has text cut off in the second column. The text in the figures are difficult to read. The display quality of the figures are poor - please increase the resolution of the figures.

2. Please include in the limitation the impact on the major assumptions in the model and how that likely affects the results.

Please be sure to include a cover letter with your revision that addresses all of the reviews and comments as well as my comments, point-by-point. Your revision should also carefully follow the journal style as presented in the Instructions for Authors available at the journal website.

Reviewers' comments:

Reviewer's Responses to Questions

**Comments to the Author**

1. Is the manuscript technically sound, and do the data support the conclusions?

Reviewer #1: Partly

Reviewer #2: Yes

2. Has the statistical analysis been performed appropriately and rigorously? 

Reviewer #1: Yes

Reviewer #2: Yes

3. Have the authors made all data underlying the findings in their manuscript fully available?

Reviewer #1: No

Reviewer #2: No

4. Is the manuscript presented in an intelligible fashion and written in standard English?

Reviewer #1: Yes

Reviewer #2: Yes

5. Review Comments to the Author

Reviewer #1: Summary of Review

This is a review of “A focused multi-state model to estimate the pediatric and adolescent HIV epidemic in Thailand, 2005 – 2025”, submitted for publication to PLOS One. The paper is a modeling study that uses a state-transition model to estimated the magnitude of the HIV epidemic among children and youth living with HIV (CYHIV) aged 0-25 years in Thailand, projecting forward from 2005 to 2025. Overall, the manuscript addresses a clearly important question in an underserved population, is well written and easy to read, but I had trouble following the methods and questions about some assumptions.

Specific Comments

Figure 1 shows the different population states and illustrates the transitions between these states. The picture it presents illustrates in my view a model structure that is perhaps too simple, and potentially not representative of the system. HIV transmission systems are highly non-linear, with feedbacks occurring at different levels of the model, and comprising network that are not random. The schematic used here, I believe, oversimplifies greatly the nature of the system. The implications of this limitation must be discussed.

Is it possible to provide an open-source exposition of the modeling tool that is used (Python codebase, Excel frontend)?

The quality of the figures in the main body of the paper is very poor on my system (I use Preview to view PDFs on a Mac).

Reviewer #2: 1 Page 2 Line 50; 37-60% should be 37 to 60%

2 As the main outcome of this paper is the mean projected number of HIV cases aged 0-25 in 2025 but there were some uncertainty, authors should also give the minimum and maximum numbers for the main outcomes.

3 Foreign labors are very common to have sex, be pregnant, and give births to their children in Thailand. The may also have HIV without knowing their status. How this would affect the results of this paper?

4 What would authors recommended to make the Thai HIV reporting system better, should be included in Discussion.

6. PLOS authors have the option to publish the peer review history of their article (what does this mean?). If published, this will include your full peer review and any attached files.

Reviewer #1: No

Reviewer #2: **Yes: **Bandit Chumworathayi

---

## [Author Response · Author response to Decision Letter 0]

5 Aug 2022

We are providing this point-by-point response, to the Editor's and reviewers' comments, and a revised manuscript in which the changes are highlighted in bold-faced text. 

Editor Comments (if provided):

Your paper was of interest to the reviewers. As you can see below, the reviewers identified both strengths and weaknesses in your paper. The reviewers noted important areas of your paper that require careful attention. From my own reading of your paper, I am in agreement with the reviewers that your paper will make an important contribution and that your paper will benefit from a minor revision. Please revise your paper in accordance with the reviews. In addition to the reviewer's comments please address the following in your revised manuscript.

1. Please review figures and tables - Table 2 has text cut off in the second column. The text in the figures are difficult to read. The display quality of the figures are poor - please increase the resolution of the figures.

We thank the editor for this comment. We have edited Table 2 accordingly and the resolution of the figures has been increased. 

2. Please include in the Limitations the impact of the major assumptions in the model and how that likely affects the results.

We thank the editor for this comment and have added the impact of the assumptions in the limitations : 

Line 454 : Third, we assumed exclusive formula feeding in all infants > 1 year born to mothers living with HIV, because breastfeeding beyond this age is rare among women living with HIV in Thailand. Given low maternal prevalence (<1%), and MTCT rate (<2%), even large relative changes in vertical transmission risk after age 1 would have minimal impact on our projections, and CYPHIV would remain a small proportion of the projected epidemic by 2025.

Line 459 : Fourth, our analysis is limited by gaps in available data for use as input parameters. In particular, by carrying most recent available inputs through to 2025 in our base case analysis, we have not simulated potentially improved care and interventions in the child and adolescent population over time including improved data reporting and completeness of data over time ; at the same time, we did not account for the impact of the COVID-19 pandemic on access to care. These conservative inputs however, provide the reader with a projection of the epidemic in a status quo scenario. As post-COVID services resume, and if access to HIV diagnosis and care improves over time, ART coverage would be higher than that estimated and number of new infections lower. Fifth, we deliberately do not model HIV transmission between individuals, due to lack of relevant data on sexual and injection networks in Thai youth ; we chose instead to develop a simpler model structure, using readily available data about yearly probabilty of HIV acquisition in each cohort, to identify the most relevant parameters and inform future data collection.

3. Have the authors made all data underlying the findings in their manuscript fully available?

We take seriously our obligation to make all data inputs and model results transparently available to readers. Primary sources cited in the manuscript and/or appendix. All model results are shown in the manuscript or appendix. We have also made the model source code freely available on Github, and added a section to the manuscript describing the process of disseminating this resource to local policymakers and training them in the use of the model. We welcome additional suggestions by the Editors on additional ways to display these inputs and results. 

Line 159 :

Model inputs, including all transition probabilities from one state to another were derived for each calendar year, age, sex, and population. Data were derived from national surveys and national projections available at https://hivhub.ddc.moph.go.th/epidemic.php ; data prior to 2010 are available upon request. Where national data were not available, we derived model inputs from clinical trials and cohort studies in Thailand or other settings (Table 1, S1 Tables A - C B, S1 Figs A and B). 

Line 214 :

Dissemination and training

The tool input sheets, output template and Python code are publicly available at https://github.com/MGH-MPEC/Thailand-Model. Furthermore, the Thai Pediatric and Adolescent HIV Modeling Working Group identified model users within their institutions who learned to use the model through a series of ten live training session.

Reviewer #1: Summary of Review

This is a review of “A focused multi-state model to estimate the pediatric and adolescent HIV epidemic in Thailand, 2005 – 2025”, submitted for publication to PLOS One. The paper is a modeling study that uses a state-transition model to estimated the magnitude of the HIV epidemic among children and youth living with HIV (CYHIV) aged 0-25 years in Thailand, projecting forward from 2005 to 2025. Overall, the manuscript addresses a clearly important question in an underserved population, is well written and easy to read, but I had trouble following the methods and questions about some assumptions.

We thank the reviewer for this comment. We have clarified many requested points throughout the manuscript, and specific comments are addressed below :

Specific Comments

1. Figure 1 shows the different population states and illustrates the transitions between these states. The picture it presents illustrates in my view a model structure that is perhaps too simple, and potentially not representative of the system. HIV transmission systems are highly non-linear, with feedbacks occurring at different levels of the model, and comprising network that are not random. The schematic used here, I believe, oversimplifies greatly the nature of the system. The implications of this limitation must be discussed.

The model is focused on the outcomes of children and adolescents living with HIV in terms of access to care. We made a deliberate decision not to directly simulate person-to-person HIV transmission, but instead to use the numbers of new infections each year as a model input, in the form of the probability pI_(j,k) (i) of acquiring non perinatal HIV in year i, at age j in population k. We have added this to the discussion.

Line 467 : Fifth, we deliberately do not model HIV transmission between individuals, due to lack of relevant data on sexual and injection networks in Thai youth ; we chose instead to develop a simpler model structure, using readily available data about yearly probabilty of HIV acquisition in each cohort, to identify the most relevant parameters and inform future data collection.

2. Is it possible to provide an open-source exposition of the modeling tool that is used (Python codebase, Excel frontend)?

We have made the model code and excel-based user interface available on Github (URL) and added this to the manuscript :

Line 214 :

Dissemination and training

The tool input sheets, output template and Python code are publicly available at https://github.com/MGH-MPEC/Thailand-Model. Furthermore, the Thai Pediatric and Adolescent HIV Modeling Working Group identified model users within their institutions who learned to use the model through a series of ten live training session.

3. The quality of the figures in the main body of the paper is very poor on my system (I use Preview to view PDFs on a Mac).

We acknowledge the resolution of the figures was poor. We have reformatted the figures accordingly.

Reviewer #2:

1. Page 2 Line 50; 37-60% should be 37 to 60%

We thank the reviewer for this comment and have made this revision to the text.

2. As the main outcome of this paper is the mean projected number of HIV cases aged 0-25 in 2025 but there were some uncertainty, authors should also give the minimum and maximum numbers for the main outcomes.

We agree wholeheartedly with the critical need to report the impact of uncertainty in model input parameters on model-projected results. To do this, we have conducted extensive univarate and multivariate sensitivity analyses. Given the large number of outcomes reported in the manuscript and appendix, we have not shown the minimum and maximum projected values in a Figures 2-3 (stacked bar graphs, as these would be difficutl to interpret visually) or Table 1 (primary results, because reporting the max and min values here might suggest that each outcome is equally likely). Instead, we have shown key sensitivity analysis results separately in Appendix Table F and Appendix Figures L-O.

Line 446 : There are several limitations to our analysis. First, our model does not track individual patient trajectories over time; however, « memory » of previous health states is not essential to the projections made in this analysis, and the simpler compartmental model structure allows for simple and transparent sensitivity analyses which can identify key parameters influencing model outputs. Second, we do not include estimates of uncertainty in our base-case model results. Instead, we conduct extensive sensitivity analyses to quantify the impact of uncertain in key parameters ; we found that our model was mainly sensitive to variations in the parameters defining theMSM population (high:low risk ratio, access to diagnosis and care) and the initial HIV incidence in 2005 in each group 

3. Foreign laborers are very common to have sex, be pregnant, and give birth to their children in Thailand. They may also have HIV without knowing their status. How this would affect the results of this paper?

We thank the reviewer for this comment. Number of newborns exposed to HIV was calculated by multiplying the estimated maternal prevalence (UNAIDS) by the number of births occuring in the country for a given year. Births to women who are foreign laborers are therefore accounted for in our model. We have added this explanation to the Appendix, page 4:

In each calendar year, the number of new perinatal infections was modeled as a function of maternal HIV prevalence among all women in Thailand, including foreign laborers, prevention of MTCT (PMTCT) coverage, PMTCT regimen, and regimen-specific transmission rates (Supplemental Table C, Supplemental Figure A) [9].

4. What would authors recommended to make the Thai HIV reporting system better, should be included in Discussion.

A key strength of our focused model is to identify data that will improve epidemic projections, in order to guide future data collection and reporting. We identifed two parameters as most influential : MSM risk ratio and MSM incidence. Targeted reporting on these parameters, for example through expansion of the already-robust Thai national HIV reporting system, would further improve projections of the number of children and youth with HIV in Thailand. For PLHIV registered in the National AIDS Program (>400,000 PLHIV), providers can categorize risk status (men who have sex with men, transgender women, youth with perinatally acquired HIV, people who inject drugs, commercial sex wokers). However, this variable is often missing. Improving completeness of the dta on risk category in the National AIDS Program for both new and currently-treated patients would be helpful for monitoring each population’s HIV cascade to ensure that indicators reach 95-95-95, providing targeted HIV services for specific population as needed, and improving data reports that can be used for modeling and projection in the future.

Line 397 : In addition to data specific to MSM, more data are also needed on TGW, PWID, and FSW to elucidate the role these risk groups play in the adolescent HIV epidemic. These populations are not mutually exclusive, and they are not consistently defined in currently available data. Additionally, the burden of drug use is high among young MSM and FSW populations and data available to date do not specifically address potential overlap between groups identifying as PWID, MSM, and FSW [47-49]. Consequently, our model may overestimate the adolescent and youth epidemic when high-risk populations overlap. Furthermore, inputs related to access to care were estimated from the National AIDS Programme (NAP) database. Although the care provider can categorize key population (MSM, PWID, FSW…) in the database, this variable is often missing. In our analyses, we made several assumptions based on data available from IBBS to stratify access to care by population. Improving the completeness of population-categorized data entry in NAP would be helpful for monitoring each population’s HIV cascade, providing HIV targeted services for specific populations as needed, and improving data report that can be used for modeling and projections in the future.

---

## [Decision Letter · Decision Letter 1]

5 Oct 2022

A focused multi-state model to estimate the pediatric and adolescent HIV epidemic in Thailand, 2005 – 2025

PONE-D-21-40026R1

Dear Dr. Desmonde,

We’re pleased to inform you that your manuscript has been judged scientifically suitable for publication and will be formally accepted for publication once it meets all outstanding technical requirements.

Kind regards,

Brian C. Zanoni, MD

Academic Editor

PLOS ONE

Additional Editor Comments (optional):

The authors have appropriately responded to the reviewer comments and have significantly improved this manuscript. It is now acceptable for publication.

Reviewers' comments:

Reviewer's Responses to Questions

**Comments to the Author**

1. If the authors have adequately addressed your comments raised in a previous round of review and you feel that this manuscript is now acceptable for publication, you may indicate that here to bypass the “Comments to the Author” section, enter your conflict of interest statement in the “Confidential to Editor” section, and submit your "Accept" recommendation.

Reviewer #1: All comments have been addressed

2. Is the manuscript technically sound, and do the data support the conclusions?

Reviewer #1: (No Response)

3. Has the statistical analysis been performed appropriately and rigorously? 

Reviewer #1: (No Response)

4. Have the authors made all data underlying the findings in their manuscript fully available?

Reviewer #1: (No Response)

5. Is the manuscript presented in an intelligible fashion and written in standard English?

Reviewer #1: (No Response)

6. Review Comments to the Author

Reviewer #1: (No Response)

7. PLOS authors have the option to publish the peer review history of their article (what does this mean?). If published, this will include your full peer review and any attached files.

Reviewer #1: No

---

## [Editor Report · Acceptance letter]

9 Nov 2022

PONE-D-21-40026R1 

A focused multi-state model to estimate the pediatric and adolescent
HIV epidemic in Thailand, 2005 – 2025 

Dear Dr. Desmonde:

I'm pleased to inform you that your manuscript has been deemed suitable for publication in PLOS ONE. Congratulations! Your manuscript is now with our production department. 

Kind regards, 

on behalf of

Dr. Brian C. Zanoni 

Academic Editor

PLOS ONE